# Soil Water Movement of Mining Waste Rock and the Effect on Plant Growth in Arid, Cold Regions of Xinjiang, China

**Zizhao Zhang** [1,2,*], **Qianli Lv** [1], **Zezhou Guo** [1], **Xuebang Huang** [1] **and Ruihua Hao** [1]

1   School of Geology and Mining Engineering, Xinjiang University, Urumqi 830046, China;
    13579843644@163.com (Q.L.); xiangerji427@163.com (Z.G.); hxb0714@163.com (X.H.);
    hrh@stu.xju.edu.cn (R.H.)
2   State Key Laboratory for Geomechanics and Deep Underground Engineering, Xinjiang University,
    Urumqi 830046, China
*   Correspondence: zhangzizhao@xju.edu.cn; Tel.: +86-136-3997-7295

**Abstract:** Understanding the water movement in reconstructed soil and its efficacy on local vegetation is critical for the ecological reclamation of mine lands. This study employed field experiments and a numerical model to investigate the water movement in reconstructed soil and evaluate the effects of mining waste rock on plant growth in an arid and cold region of Xinjiang. Water contents and matrix potentials were monitored over 1-year period. A numerical model was established based on the observed data to calculate soil water balance and irrigation demand. The results show that the soil water content at a shallow depth could be more vulnerable to the climate variability in uncompacted and compacted soil. The water content at the depth of 50 cm with 30 cm-thick covering soil was the lowest; meanwhile, the barrels with 50 cm- and 70 cm-thick covering soil without compaction had the highest water content. Moreover, the water content of the uncompacted soil could be lower than that of the counterpart attributed to the variation in soil porosity. To maintain the water content as an optimized value to grow a certain plant species in the long run, irrigation could be implemented according to the water balance over time in mine lands.

**Keywords:** mine lands reclamation; reconstructed soil; water movement; plant growth; arid and cold regions





## 1. Introduction

Mineral resource exploitation serves as the primary industry worldwide and it greatly promotes socioeconomic development because of the abundant reserves in resources [1,2]. However, over the past few decades, the environment has been experienced substantial damage, such as the barrenness of land resources and destruction of the soil and vegetation on the land surface, due to large-scale exploitation [3–5]. Because of the extreme climate and barren water and soil resources, the ecological environment in arid and cold areas of Xinjiang is in particular vulnerable to mining activity.

To restore the damaged soil and the ecological environment, efficient methods have been employed to reclaim the mine lands [6]. Land reclamation measures for open-pit mining and ground collapse are frequently employed and the procedures are as follows: first, waste rock backfill is applied for landform restoration; second, the surface of the waste rock is covered with soil; third, vegetation is planted on the surface of the waste rock to restore the ecological environment. Since soil and water resources are extremely scarce in this region [3,6], it is necessary to optimize the soil covering scheme during land reclamation to survive the vegetation and guarantee its growth. As such, it is important to investigate soil water movement in reconstructed soils and evaluate its ecological efficacy. The factors affecting water movement in reconstructed soil have been studied, and a couple of methods have been applied to analyze reconstructed soil water characteristics in mining areas. Column experiments were used to study water movement characteristics in covering

soil filled with gangue and fly ash and these studies found that water movement could be affected by the relations of particles makeup between the soil and the filling, porosity, and water-holding capacity [7–10]. Whalley [11] applied time-domain reflectometry (TDR) to measure soil water content and developed a simple linear calibration function for water balance.

In mining areas, research on reconstructed soil water movement has primarily been conducted under conditions where the lower layer of the soil is backfilled with fly ash and coal gangue. For example, Younger et al. [12] found that the water content and bulk density of the reconstructed soil were higher than those of the natural soil; meanwhile, the infiltration rate of the reconstructed soil was lower. Research results [13–16] have shown that when using fly ash as the filling matrix for reconstructed soil, a difference between the capacity for water movement and natural soil may exist. Gerke et al. [17] simulated the water movement characteristics of reconstructed soil pores filled with coal gangue embedded in lignite particles, and found that for unsaturated flow conditions at higher matric potential heads, water in a restricted part of the fragment domain must be more mobile as compared to water in the sandy matrix domain. Lu [18] studied the effect of compaction on crop yield and soil water in the upper layer of a coal gangue filling matrix, and found that the soil moisture in the surface layer would be increased with the compaction layer thickness and there would be an obvious positive relation between the crop yield and the moisture content of the surface layer soil. Other researchers [19–21] have analyzed the influence of different thicknesses and locations of coal gangue on the process of soil water infiltration.

Although the reconstructed soil water movement and its ecological effects in mining regions backfilled with coal ash and coal gangue have been studied, to date and to our knowledge, water movement in sandy soil with waste rock filling and its effect on plant growth under arid and cold climate conditions are still knowledge gaps. The porosity, hydraulic conductivity, and other hydraulic features of the lumpy waste rock are significantly different from coal ash and coal gangue. The objective of this study is to investigate the water movement in sandy soil with waste rock filling and evaluate the effects of the reconstructed soil with different covering thickness on a plant growth under arid and cold climate conditions. A field test and long-term dynamic monitoring were applied to determine the variability of the water content in reconstructed surface soil over a 1-year irrigation period. Based on the observation data, soil covering scheme was optimized for the growth of a certain plant species and a numerical model was established to evaluate the irrigation demand. This study seeks to answer the following questions: (1) What are the water contents in different soil thickness levels with and without compaction filled with waste rock under an arid and cold climate? (2) How does the reconstructed soil with a different covering depth affect the growth of a certain plant species under an arid and cold climate?

## 2. Materials and Methods

This section is organized as follows: (1) description of the study site; (2) field test; and (3) numerical modeling.

### 2.1. Study Area

The study site, the Changji groundwater test field, is in the Beimen village, Changji prefecture, Xinjiang province, China (Figure 1). The site is located 2 km north of the Changji downtown area and 40 km west of Urumqi City (Figure 1). The climate is typical continental arid with hot summers and cold winters. The average annual temperature, precipitation, and evaporation are 7.5 °C, 122.8 mm, and 1266.8 mm (The data were obtained from a meteorological station, which is used to measure the ground-based meteorological data including precipitation, evaporation, wind direction and speed, and temperature). The study site is equipped with three kinds of test containers. A gallery-style underground

observation room equipped with the ground osmometer observation system and a shaft-type negative pressure observation room have been established.

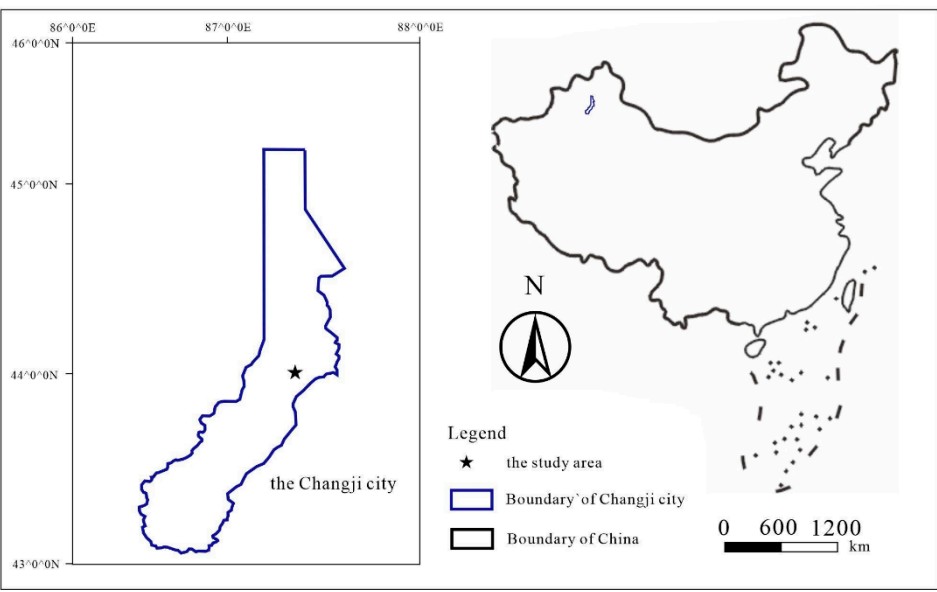

**Figure 1.** Study site: Changji groundwater balance test field.

*2.2. Field Test*

A field test was conducted at the study site to investigate the water content in reclamation soil, evaluate its effects on the growth of a certain plant species, and provide observation data to build a numerical model. The field test progressed sequentially beginning with sampling of the soil and waste rock and property characterization of the samples, then progressing to a barrel experiment. The test soil and waste rock were sampled from a mine land (located in Dabancheng, Urumqi, Xinjiang, approximately 100 km away from the study site) which is currently under exploitation and under the same climate conditions. The main soil type in the study area is sandy soil, brown calcareous soil, chestnut calcareous soil, and a small amount of desert soil. The brown calcareous soil was sampled because it is mainly distributed in the piedmont alluvial fan at the northern Tianshan Mountain where the study site is located. The surface layer of the soil is mainly composed of vegetation humus caused by long-term humus and calcareous activities. The underlying layer gradually reveals a light white calcium deposit. Property characterization of the soil and waste rock was conducted in the lab in succession with sample collection. Specifically, particle analysis and leaching test were implemented to obtain the granulometry and mineralogy of soil samples.

A total of 6 test barrels were used for the experiment (Figure 2). The surface area and height of each barrel are 2 m² and 2.2 m, respectively (Figure 2). The barrels were divided into three groups with surface soil thicknesses of 30 cm (a in Figure 2), 50 cm (b in Figure 2), and 70 cm (c in Figure 2). In each group, the soil in one of the barrels was not compacted; however, the soil was compacted once using a bulldozer in the other one. The neutron probe and tensiometers were used to determine the water content and matrix potential of the reconstructed soil at different depths, respectively. The neutron probe can be used to measure the soil moisture of an entire soil layer or profile, which can provide efficient and accurate monitoring data. Since the aluminum tube which had been originally equipped with the neuron probe could not be qualified by the experiment, it was substituted for by the iron tube and stainless steel tube. The two tubes were both preplaced in the six barrels to fix the neuron probe. To compare the results obtained from different tubes, an aluminum tube was also preplaced in one of the barrels (i.e., c2 in Figure 2). Neutron probes were respectively fixed in the tubes at depths of 10 cm, 30 cm, 50 cm, 70 cm, and 90 cm in each barrel. Tensiometers were respectively installed at depths of 10 cm and 30 cm for the

30 cm-thick covering soil; for the 70 cm-thick covering soil, tensiometers were installed at 10, 30, and 50 cm, respectively; similarly, tensiometers were installed at 10, 30, 50, and 70 cm for the 70 cm-thick covering soil. The arrangements of tensiometers and neutron probes are shown in Figure 2B.

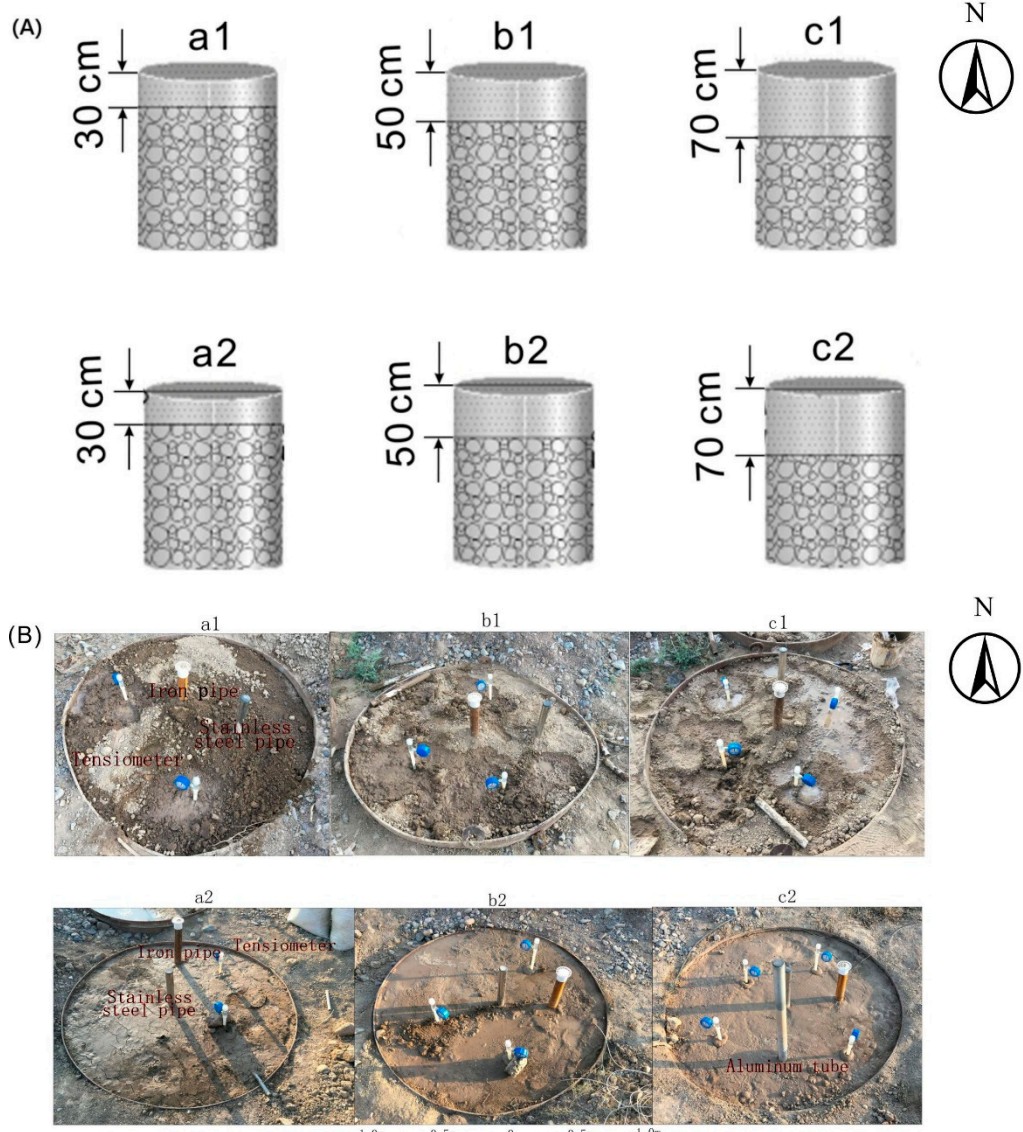

**Figure 2.** The 6 barrels applied in the field test. (**A**) Front view. (**B**) Plan view. Note: a, b, and c represent the covering soil with thickness of 30 cm, 50 cm, and 70 cm, respectively; 1 and 2 representing the topsoil are not compacted and one-time compacted, respectively. For example, a1 represents the 10 cm-thick covering soil without compaction.

In the experiment, irrigation frequency was assessed consistent with that implemented by an artificial grassland in this area (3–5 times/year). The irrigation amount employed in the experiment was quantified based on the amount used in the mine lands in Xinjiang [240 m$^3$/hm$^2$] and surface area of each test barrel. As such, 28.8 L, 48 L, and 67.2 L of water were respectively applied for the test barrels: (1) 30 cm-thick covering soil (equivalent to 14.4 mm precipitation), (2) 50 cm-thick covering soil (equivalent to 24 mm precipitation), and (3) 70 cm-thick covering soil (equivalent to 33.6 mm precipitation). Irrigations were implemented on 7 August, 8 September, and 4 October, 2016, respectively. The monitoring started on 5 August 2016. Since quick variations in soil water content can take place in the beginning of the experiment to have a better understanding of this pattern, daily

observations were conducted in the first month after the beginning of the test. Thereafter, interval observations (2–4 times per week) were initiated. In certain circumstances such as precipitation and irrigation events, the observation frequency was increased correspondingly. Because the temperature decreases sharply in the study area at the end of October, to prevent the tensiometers from being damaged by the harsh climate, the monitoring using tensiometers was stopped from 31 October 2016 to 22 April 2017; during this period, only the observation data from the neutron probes were available. The experiment had been conducted for one year. The field test data were processed and are shown in the results section.

### 2.3. Numerical Modeling

2.3.1. Water Movement in Reconstructed Soils

To simulate vertical movement of water flow in reconstructed soils, a one-dimensional conceptual hydrogeological model was created based on the field test settings, which is shown in Figure 3A. The model is represented by a one-dimensional saturated-unsaturated Richards equation [22]:

$$\frac{\partial \theta}{\partial t} = \frac{\partial}{\partial z}(K\frac{\partial h}{\partial z}) + \frac{\partial}{\partial z}K(s) \tag{1}$$

where $K$ is hydraulic conductivity (L/T), $\theta$ is the mean soil volumetric water content ($L^3/L^3$), and $h$ is the matrix potential.

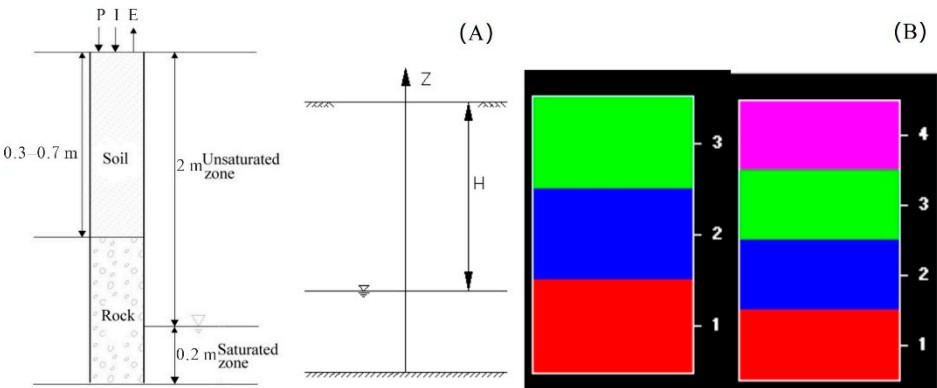

**Figure 3.** (**A**) The conceptual model to simulate soil water movement. (**B**) The test barrels in the modeling.

The soil water characteristic curve (SWCC) is the most basic hydraulic characteristic curve for solving the equation of soil-water flow movement. The van Genuchten equation for describing the soil water characteristics is shown as follows [23]:

$$\theta(h) = \begin{cases} \theta_r + \frac{\theta_s - \theta_r}{[1+|\alpha h|^n]^m}, h \leq 0 \\ \theta_s, h>0 \end{cases} \tag{2}$$

where $\theta(h)$ is the volumetric moisture content; $\theta_r$ and $\theta_s$ are the residual water content and saturated soil water content ($L^3/L^3$); m and n are the parameters of the model, respectively; $\alpha$ (>0, $L^{-1}$) is the reciprocal of the air entry value; and $n$ (>1) is the number of parameters that affect the shape of the water characteristic curve ($m = 1 - 1/n$).

The initial condition for simulating water content was set as the beginning of the observation (5 August 2016). The upper boundary condition was determined by meteorological conditions. Based on the meteorological data, surface evaporation was set as the upper boundary condition. The constant head boundary was set as the lower boundary.

The total thickness of the model was set as 220 cm (Figure 3B), which was consistent with that in the mine land where the soil and waste rock were sampled (Section 2.2). In the model, based on the particle analysis from the field test (Section 2.2), the test barrels a and

b (c) were divided into three or four layers (Figure 3B). A significant difference in particle sizes between the soil and waste rock have the hydraulic parameters changed dramatically during the movement of water, making the numerical simulations difficult to converge. Therefore, the interface between the soil and waste rock was divided as one or two layer(s).

The model was numerically solved by the water flow calculation module in HYDRUS-1D [24]. In total, 82 daily observed data (5 August 2016–25 October 2016) from field tests were used for the numerical modeling. The initial time step was set as 0.001 days, which then was automatically adjusted by the program based on the computing convergence criterion. The maximum and minimum time step was 1 day and $1 \times 10^{-5}$ days, respectively. The model calibration parameters are shown in Table 1.

**Table 1.** Hydraulic soil water parameters.

| Number | Depth (cm) | $\theta_r$ (cm$^3$/cm$^3$) | $\theta_s$ (cm$^3$/cm$^3$) | $\alpha$ (cm$^{-1}$) | $n$ (−) | $k_s$ (cm·d$^{-1}$) | $l$ |
|---|---|---|---|---|---|---|---|
| a1 | 0–20 | 0.01 | 0.25 | 0.020 | 1.960 | 250 | 0.5 |
| | 20–40 | 0.001 | 0.51 | 0.050 | 1.710 | 300 | 0.5 |
| | 40–200 | 0.001 | 0.50 | 1.083 | 1.413 | 7994 | 0.5 |
| b1 | 0–40 | 0.01 | 0.44 | 0.018 | 6.080 | 250 | 0.5 |
| | 40–60 | 0.001 | 0.51 | 0.065 | 1.677 | 300 | 0.5 |
| | 60–200 | 0.001 | 0.50 | 2.754 | 1.219 | 7994 | 0.5 |
| c1 | 0–40 | 0.01 | 0.36 | 0.013 | 9.070 | 200 | 0.5 |
| | 40–60 | 0.01 | 0.36 | 0.018 | 10.000 | 100 | 0.5 |
| | 60–80 | 0.001 | 0.51 | 0.073 | 1.922 | 300 | 0.5 |
| | 80–200 | 0.001 | 0.50 | 3.244 | 1.344 | 8000 | 0.5 |
| a2 | 0–20 | 0.01 | 0.21 | 0.030 | 1.184 | 10 | 0.5 |
| | 20–40 | 0.008 | 0.41 | 0.059 | 1.452 | 100 | 0.5 |
| | 40–200 | 0.001 | 0.51 | 1.158 | 1.382 | 7697 | 0.5 |
| b2 | 0–40 | 0.01 | 0.32 | 0.011 | 10.000 | 29 | 0.5 |
| | 40–60 | 0.001 | 0.49 | 0.031 | 2.945 | 482 | 0.5 |
| | 60–200 | 0.001 | 0.60 | 3.707 | 1.125 | 10,817 | 0.5 |
| c2 | 0–40 | 0.01 | 0.60 | 0.018 | 4.250 | 500 | 0.5 |
| | 40–60 | 0.01 | 0.40 | 0.026 | 2.634 | 500 | 0.5 |
| | 60–80 | 0.008 | 0.41 | 0.124 | 1.542 | 100 | 0.5 |
| | 80–200 | 0.001 | 0.51 | 1.000 | 1.417 | 7697 | 0.5 |

### 2.3.2. Irrigation Demand for Land Reclamation

The irrigation demand for land reclamation was simulated based on the data from field tests and the inversions of the numerical model described above. In total, 20-years of meteorological data from the Changji meteorological station over the nonfrozen period (each year from 1 April to 1 October, i.e., 214 days/year) were used for the modeling.

HYDRUS-1D was also employed to solve the irrigation demand for land reclamation. The initial condition of the model was set as the field water capacity. The upper boundary condition was the meteorological boundary (the second type of boundary condition), and the fixed water level was set as the lower boundary. Model resolution was set as 1 cm. The initial time step was set as 0.1 day. The minimum and maximum time step was $1 \times 10^{-4}$ days and 0.1 days, respectively.

## 3. Results

### 3.1. Field Test Data

The properties of soil samples were measured in the lab to investigate its physical and chemical properties. The soil was brown calcium sandy with a bulk density of 12.646 kN/m$^3$, porosity of 57.71%, and volumetric water content of 11.76%. In soil samples, a particle size smaller than 0.5 mm accounted for 63.7% of the sample. Nonuniform particle sizes existed in the waste rock, most of which were larger than 5 cm. CaO was the dominant

component in the waste rock and the content could reach as high as 53.8%. $SiO_2$, $MgO$, and $Al_2O_3$ were the additional components that composed the rock.

In total, 2424 sets of soil moisture data and 1368 sets of matrix potential data were obtained from neutron probes and tensiometers over the period 5 August 2016 to 5 August 2017. Comparisons were made between the results obtained from the neutron probe fixed in the substitute tubes (iron and stainless steel) and the original (aluminum) tube. Table 2 shows the data correlations between the substitute tubes and original tube based on the 360 sets of data acquired immediately after the initiation of monitoring using Bivariate Correlations methods. Significant correlations took place between the data from neutron probe fixed in the three types tubes; however, a higher correlation between the data from the stainless steel tube and the aluminum tube was presented (Table 2). Therefore, the neutron probe data from the stainless steel tube were used for the calculation of volumetric soil water content. Volumetric water content was converted based on calibration equations (Table 3) for the soil layer profile, which was established by relying on the experimental research results of the test field.

**Table 2.** Correlation analysis of neutron soil moisture gauge data obtained from stainless steel, iron, and aluminum tubes.

|  |  | Stainless Steel Tube | Aluminum Tube | Iron Tube |
|---|---|---|---|---|
| Stainless steel tube | Pearson correlation | 1 | 0.939 [a] | 0.928 [a] |
|  | significance (two-tailed) |  | 0.000 | 0.000 |
|  | N | 360 | 360 | 360 |
| Aluminum tube | Pearson correlation | 0.939 [a] | 1 | 0.899 [a] |
|  | Significance (two-tailed) | 0.000 |  | 0.000 |
|  | N | 360 | 360 | 360 |
| Iron tube | Pearson correlation | 0.928 [a] | 0.899 [a] | 1 |
|  | Significance (two-tailed) | 0.000 | 0.000 |  |
|  | N | 360 | 360 | 360 |

[a] The two-tailed test level of significance is $p < 0.01$.

**Table 3.** Calibration equations for neutron probe data in the soil.

| Serial Number | Instrument Model of Neutron Probe | Depth of Measurement (cm) | Calibration Equation |
|---|---|---|---|
| 1 |  | 10 | $\theta$ [a] $= 7.12 + 49.61 \times (R$ [b] $/R\omega$ [c]$)$ |
| 2 | L-520D | 30 | $\theta = 5.071 + 58.224 \times (R/R\omega)$ |
| 3 |  | $\geq$40 | $\theta = -0.83 + 58.102 \times (R/R\omega)$ |

[a] Volumetric soil water content (%). [b] Average of neutron readings at different depths of soil (counts/s). [c] Calibration number of neutron probe (counts/s).

### 3.2. Soil Water Content and the Effect on a Plant Growth

Figure 4 shows the variations in water content in the reconstructed soil with different covering thickness with or without compaction during 5 August 2016–5 August 2017. The variations in water content in different thickness of soil showed similar patterns but different magnitudes. As described in Section 2, irrigations were implemented monthly from August to October, which may supplement recharge during the arid period. Therefore, water contents in different thickness of soil could maintain stable levels even though fluctuations took place. From the beginning of the 2017 to the spring, snow melt infiltrated the soil, so the water contents could be further stabilized during this period. Thereafter, intense evapotranspiration led to successive decreases in soil water contents to the end of study period.

Because the reconstructed soil could be easily affected by the climate, e.g., low-frequency precipitation and intense evapotranspiration, the water content was lowest at the depth of 10 cm and minor differences in the water content at this depth took place in each covering soil scenario. Further, the water content was increased with the soil depth. As shown in Figure 4, in comparison to the depth of 10 cm, the water content was

higher at the depth of 30 cm in each barrel. Besides the decreased influence of arid climate to the deeper soil, the porosity might also play an important role to lead to the results. Zhang et al. [25] analyzed the physical property of the reconstructed soil and showed that at an early stage, the fine particle content in the soil at the depth of 30 cm was generally lower than that at other soil depths. Zhang et al. [26] also presented that the porosity of the soil at the depth of 30 cm was generally larger, which could provide more space for the storage of water.

There are several indications about the effect of reconstructed soil on a plant growth that can be provided by the variations in soil water content. The growth and development of plant roots are closely related to the variation of soil moisture. In the early stage, a minor decrease in water content could particularly promote the lateral growth of plant roots [27]. In addition, the absorption position of the root system could change with the water movement in different soil depths [27]. *Medicago sativa* is frequently used to reclaim nonmetal mine lands in Xinjiang. Therefore, the effects of reconstructed soil on this plant growth are significant for restoring local environments [28]. Its root length can be varied with its growth stage: it could be approximately 10 cm at the early growth stage of the plant, 50 cm in 6 months [12–14], and more than 1 m after 3–4 years' growth [29]. First, the variations in soil water content with different covering thickness analyzed in this study could provide indications about the optimum thickness of covering soil to plant *Medicago sativa*. As shown in Figure 4, the variations in soil water content at the depth of 10 cm (corresponding to the early growth stage of *Medicago sativa*) are insignificant; meanwhile, the water content in the barrel with 50 cm covering soil (barrel b1) was the highest. Figure 5A shows the comparisons of the variations in water content at the depth of 50 cm (corresponding to the half-year growth stage of *Medicago sativa*) in the 6 barrels. It can be observed that the water content at the depth of 50 cm with 30 cm-thick covering soil (barrel a1) was the lowest, whereas the barrels with 50 cm- and 70 cm-thick covering soil without compaction (barrels b1 and c1) had the highest water content at the depth of 50 cm. Therefore, a minimum of 50 cm-thick covering soil is recommended to plant *Medicago sativa*, since a higher water content might become available. Further, as shown in Figure 5B, the water content of uncompacted soil decreased more sharply over time compared with that of the compacted soil, suggesting the water content of the uncompacted soil may be even lower than that of its counterpart over a long period. As such, as the root of *Medicago sativa* can grow to be as deep as 1 m [29], the once-compacted soil may contribute more to the growth of the plant for the long run.

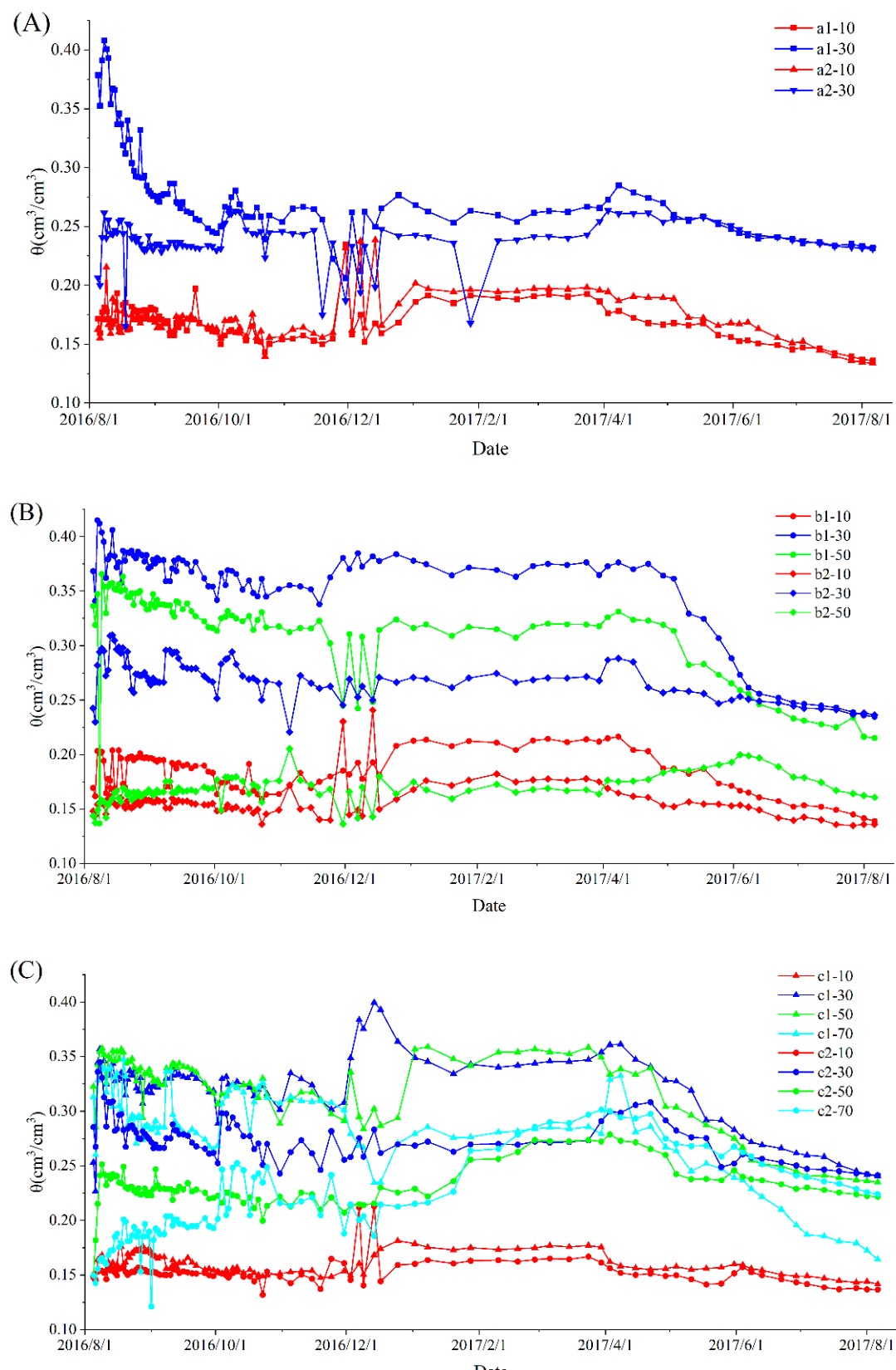

**Figure 4.** Time series of the water content at (**A**) 30 cm-, (**B**) 50 cm- and (**C**) 70 cm-thick covering soil.

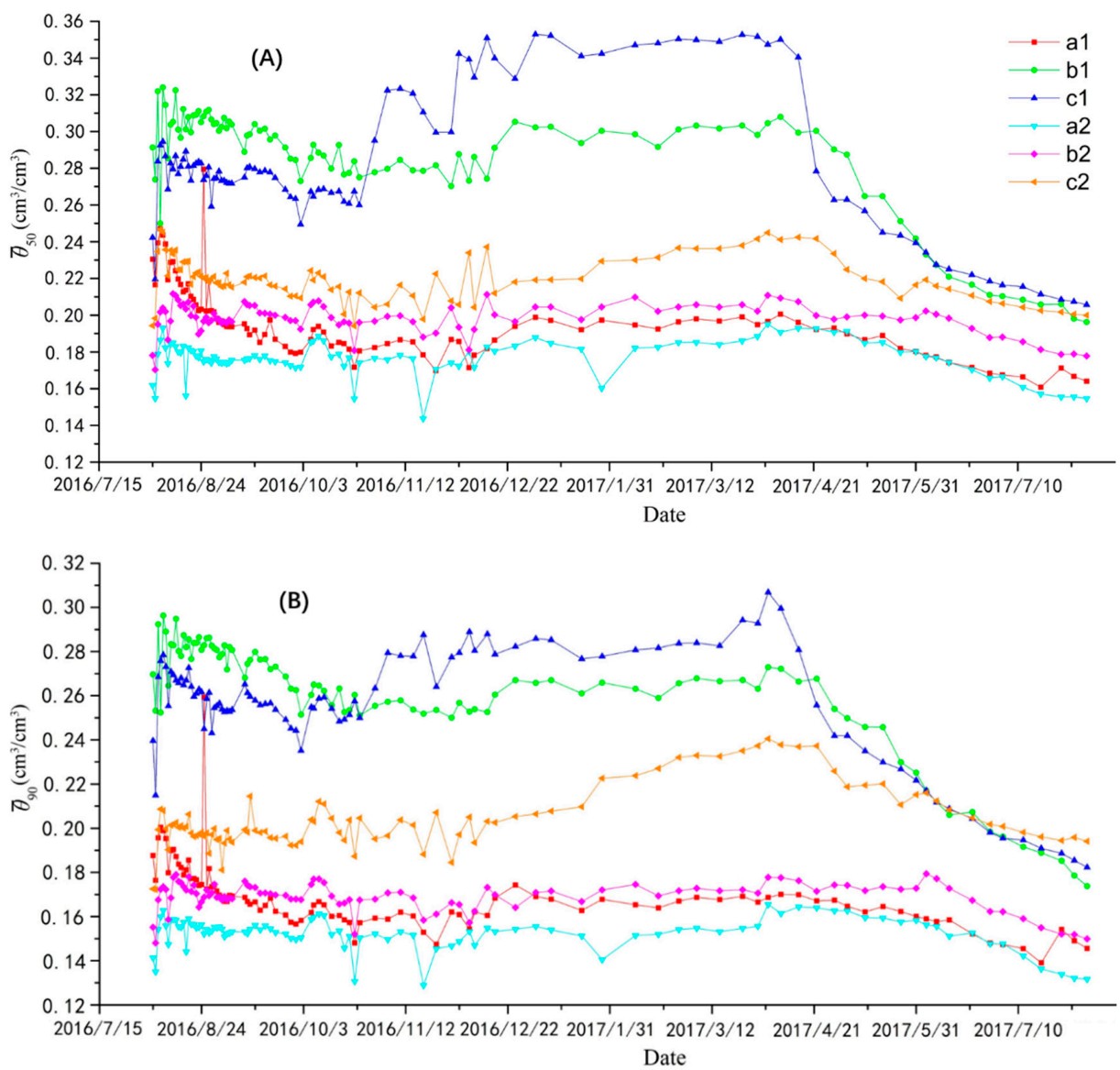

**Figure 5.** Time series of the water content at the depth of (**A**) 50 cm and (**B**) 90 cm in reconstructed soil with various covering thickness in each test barrel.

### 3.3. Water Balance Analysis

The mean RMSE between the observations and simulate results is 0.0475 cm³. Figure 6 shows part of the soil moisture content and temperature obtained from numerical calculation and observed. It can be see the calculated values fit well with the observed values, and the numerical model basically reflects the change of soil moisture content in the reconstructed soil. The initial parameters of soil water movement by the numerical model are presented in Table 3, and the variations in infiltration, evaporation, bottom flux, and soil water content for the 30 cm-thick covering soil obtained from the numerical model are shown in Figure 7A. Compared to the results from the field test (Figures 4 and 5), decreasing trends of soil water content from the numerical model also occurred for both compacted and uncompacted soils. The water content for uncompacted soil was lower than that for the counterpart. As more water was stored in the uncompacted soil, the cumulative infiltration therein was correspondingly lower than that of the compacted soil. The differences in evaporation and bottom flux between the uncompacted and compacted soils were both insignificant.

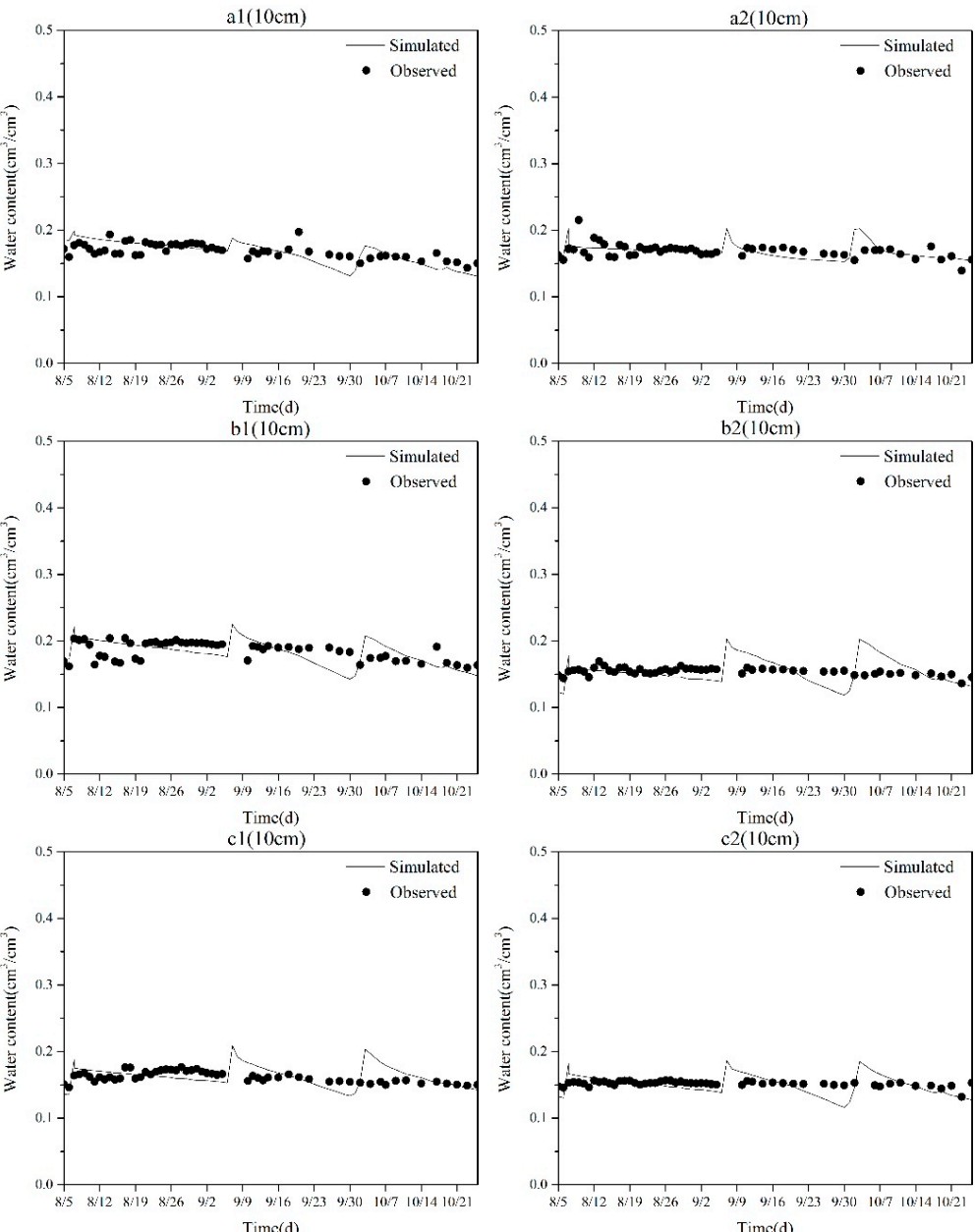

**Figure 6.** Comparison of observed and calculated soil water contents at 6 barrels in the depth of 10 cm.

The water contents in the 50 cm-thick and 70 cm-thick soils also present decreasing trends (Figure 7B,C), which are consistent with the results obtained from the field tests (Figures 4 and 5). As discussed earlier, the compaction compressed the space among the soil particles, which impeded the water evaporated from the soil, thus providing a medium to store more water compared to the uncompacted soil. Therefore, as shown in Figure 7B, for the 50 cm-thick covering soil, the water loss via evaporation for the compacted scenario was less than that for the uncompacted one. Similar to the 30 cm-thick covering soil, the differences in evaporation and bottom flux for both the 50 cm- and 70 cm-thick soils were insignificant.

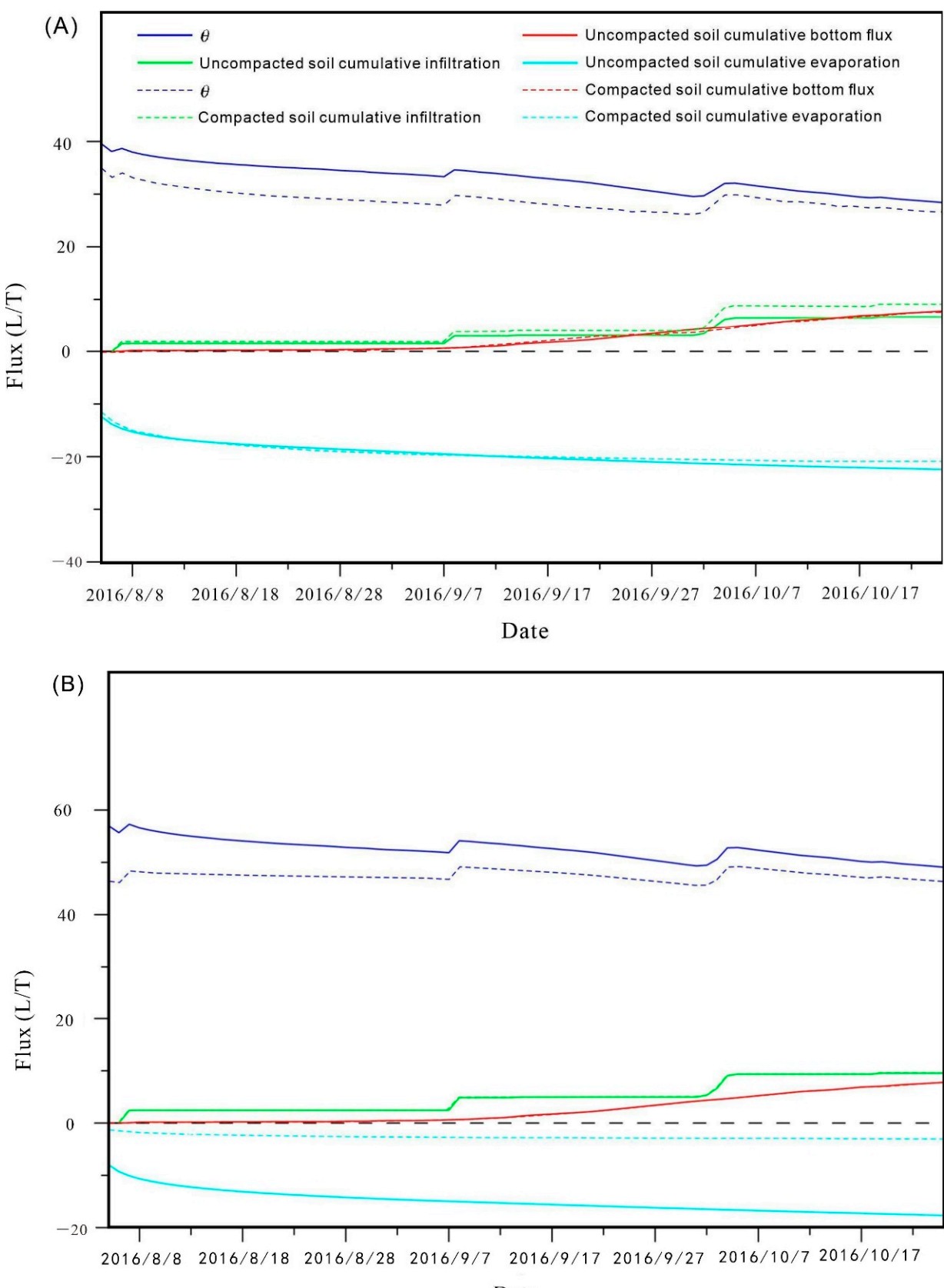

**Figure 7.** *Cont.*

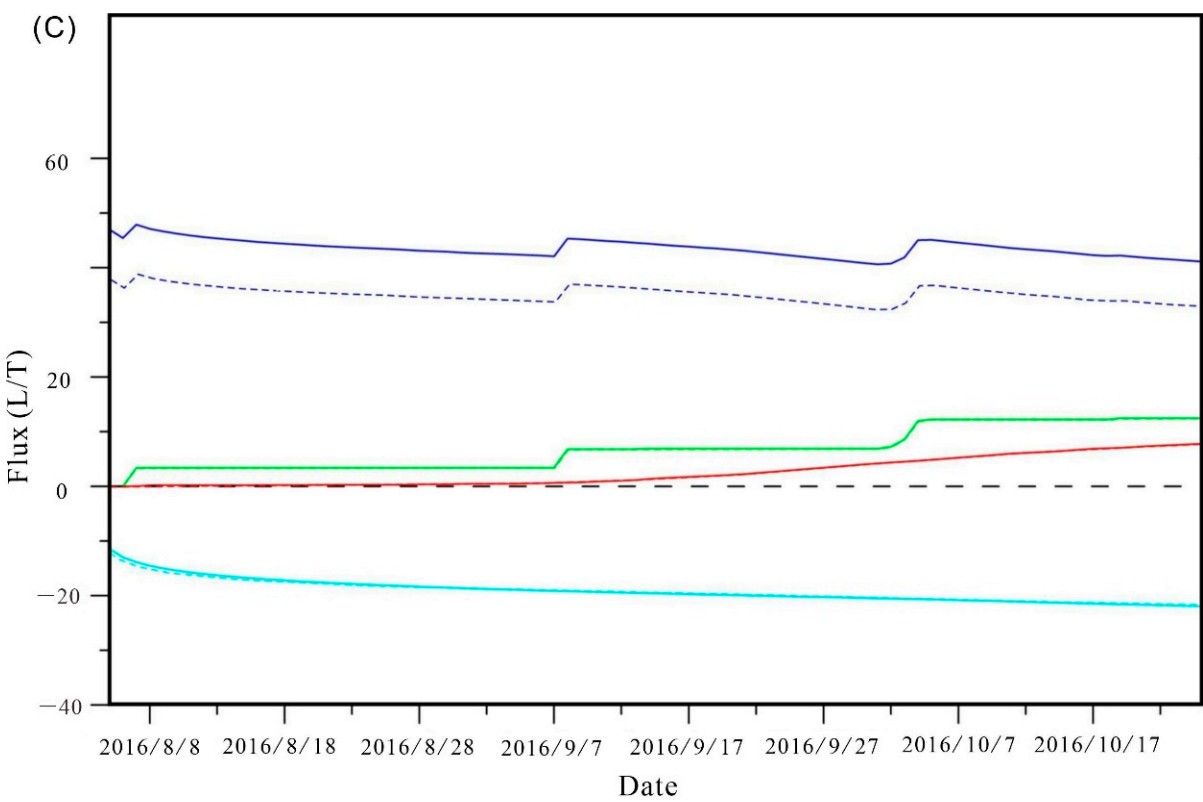

**Figure 7.** Water balance analysis for (**A**) 30 cm-, (**B**) 50 cm-, and (**C**) 70 cm-thick covering soil over the study period.

## 4. Discussion

Based on the results from field tests and the inversions of the numerical model, the irrigation demand was simulated based on the inversion parameters for the 50 cm-thick compacted soil. The initial parameters of soil water movement by the numerical model are presented in Table 4. The variation in water balance from the numerical simulation is shown in Figure 8. Both the irrigation and evaporation have linear increments with time. The simulation indicates that the primary supplement to water balance for 50 cm-thick covering soil with one-time compaction is from irrigation (precipitation is scarce in the study area, see Section 2), e.g., 286 cm at the end of the period (31 October 2018), and the main consumption of water is evaporation, e.g., 296 cm at the end of the period. The variation in cumulative soil water content was 2.18 cm, whereas the cumulative bottom flux was 8.8 cm. Compared to the irrigation and evaporation, the variations in soil water volume and cumulative bottom flux were insignificant, which may pose minor impacts on water balance.

**Table 4.** Initial hydraulics soil water parameters.

|  | Depth (cm) | $\theta_r$ (cm$^3$ cm$^{-3}$) | $\theta_s$ (cm$^3$ cm$^{-3}$) | $\alpha$ (cm$^{-1}$) | $n$ (–) | $K_s$ (cm d$^{-1}$) |
|---|---|---|---|---|---|---|
| | 0–20 | 0.034 | 0.46 | 0.016 | 1.37 | 6 |
| a1, a2 | 20–40 | 0.078 | 0.43 | 0.036 | 1.56 | 24.96 |
| | 40–200 | 0.045 | 0.43 | 0.145 | 2.68 | 712.8 |
| | 0–40 | 0.034 | 0.46 | 0.016 | 1.37 | 6 |
| b1, b2 | 40–60 | 0.078 | 0.43 | 0.036 | 1.56 | 24.96 |
| | 60–200 | 0.045 | 0.43 | 0.145 | 2.68 | 712.8 |
| | 0–40 | 0.034 | 0.46 | 0.016 | 1.37 | 6 |
| c1, c2 | 40–60 | 0.031 | 0.41 | 0.015 | 1.5 | 4 |
| | 60–80 | 0.078 | 0.43 | 0.036 | 1.56 | 24.96 |
| | 80–200 | 0.045 | 0.43 | 0.145 | 2.68 | 712.8 |

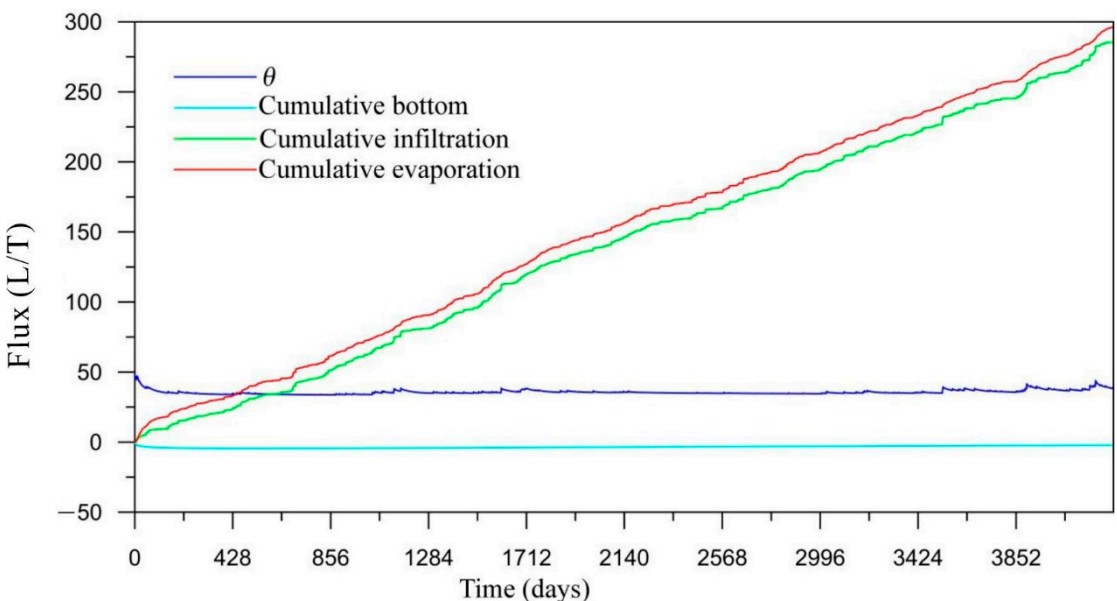

**Figure 8.** Water balance analysis for 50 cm-thick covering soil with one-time compaction over a 20-year non-freezing period.

In Figure 8, the irrigation and evaporation values are basically equivalent, indicating that the water loss from evaporation after land reclamation could be largely compensated for by irrigation. The results also imply that to maintain the water content in the soil as an optimized value to grow the plant in the long run, irrigation can be implemented according to the water balance over time analyzed by the modeling. *Medicago sativa* served as the plant to reclaim the nonmetal mine lands in the study area. Specific plant species may be employed in other unrecovered mine lands with similar climates. The simulation discussed herein may serve as an additional guidance to support irrigation policy to optimize the plant growth that reclaims the mine lands.

## 5. Conclusions

In this study, field tests and a numerical model were established and employed to investigate water movement in uncompacted and compacted reconstructed soil with different thicknesses and evaluate its efficacy on a plant growth. Given the field test data, soil water contents in reconstructed soil with different covering thickness were analyzed, and the effect of the reconstructed oil on the growth of a certain plant species were evaluated. Based on the inversion parameters, the irrigation demand over a long period for the 50 cm-thick compacted soil was simulated. The results show that the soil water content at shallow depths can be more vulnerable to be affected by climate in both uncompacted and compacted soil due to vulnerability to the climate. Moreover, the water content at the depth of 50 cm with 30 cm-thick covering soil was the lowest, whereas the barrels with 50 cm- and 70 cm-thick covering soil without compaction had the highest water content at the depth of 50 cm. Additionally, the water content of the uncompacted soil may be even lower than that of the counterpart over a long period. These results may provide an optimum thickness of reconstructed soil to grow a certain plant species that reclaims mine lands. Finally, the results imply that to maintain the water content in the soil as an optimized value to grow a certain plant species in the long run, irrigation could be implemented according to the water balance over time.

This research is the first application of field experiment and numerical model to analyze soil water movement and evaluate its effect on a plant growth in cold and arid regions, which may serve as an additional guidance to support the irrigation policy to optimize plant growth that reclaims mine lands in areas with similar climates. The limitations of

this study include: the porosity and other soil characteristics (e.g., bulk density) were not measured and analyzed between compacted and uncompacted soils, and the ecological effects of reconstructed soil were not deeply studied. Future research could focus on the evaluation of the ecological effect of the reconstructed soil, and investigate the how porosity and depth varied in compacted and uncompacted soil.

**Author Contributions:** Conceptualization, Z.Z.; methodology, Z.Z.; validation, Q.L., Z.G., X.H., and R.H.; formal analysis, Z.Z.; investigation, Z.G., R.H.; resources, X.H.; data curation, Q.L.; writing—original draft preparation, Z.Z.; writing—review and editing, Q.L.; visualization, Q.L.; supervision, Z.Z.; project administration, Z.Z.; funding acquisition, Z.Z. All authors have read and agreed to the published version of the manuscript.

**Funding:** This research was funded by the National Natural Science Foundation of China, grant number 41967036, and the Foundation for the Start-up Project of Doctoral Research of Xinjiang University.

**Data Availability Statement:** The data used to support the findings of this study are included within the manuscript.

**Conflicts of Interest:** The authors declare that there are no conflict of interest.

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
