# Peer review of "Soil Water Movement of Mining Waste Rock and the Effect on Plant Growth in Arid, Cold Regions of Xinjiang, China"

_water, doi:10.3390/w13091240_

Round 1

Reviewer 1 Report

This study performed field experiment and numerical model to evaluate water movement in soil according to different thickness of soil and compaction for the soil reclamation of mine lands. The study topic is interesting to suggest soil reconstruction condition appropriate for a plant growth to reclaim mine lands. But, there are major things to be improved to enhance the quality of this study.

  • One of the main objectives of this study is to evaluate ecological effect of soil reclamation in mine lands. In my opinion, the author seemed to exaggerate their study results because they simply explained about soil water content for a plant growth. Moreover, this study discussed very limited information of the soil water content (just provided maximum soil water content measured in the field did not considering adequate soil water content for growing Medicago sativa). Very weak results are provided in the manuscript that irrigation is needed for preventing water loss by evaporation. For maintaining this topic in the manuscript, more and reliable results related to the ecologic effect should be included.

  • A numerical model was conducted to estimate water balance in the experimental soil columns, which is an important discussion of this study. To present simulated results for the real condition, a model calibration should be completed, which was not considered in this study. So, the author is needed to provide the model calibration result (using the measured soil water content) in the manuscript for enhance reliability in their model.

  • It is better to reorganize description about the soil material in L.109~119 into result part. Additionally, information of difference in soil parameters (porosity, volumetric water content, bulk density) between uncompacted and once-compacted soil should be provided in the manuscript because this study emphasized soil water characteristics depending on the reconstructed soil condition (difference in soil compaction).

Reviewer 2 Report

The paper addresses an interesting topic. i found the introduction to be rather long as it contains a part of the literature review. I would suggest the authors to divide it intro introduction and literature review. Having a separate section of literature review, the authors can present more in depth the results of the selected papers, better shaping the gap in the literature and the novelty of the paper.

For the part related to the field test I think that the authors could try to present it as a methodology giving first some general steps to be considered and then showing how the authors have used them in a real-life situation. Some general steps would be helpful for other authors who might decide to reproduce the study in similar conditions.

In the abstract the authors mention that the water has been monitored for 1 year period. In page 2, row 190, it is stated that there have been 82 daily observed data, while in row 211 appears that the the period is of 1 year. Can you please better state why and how you have decided to use the 82 days extracted data. This part is not clearly stated in the paper.

Please discuss the results in comparison with other studies from the field and please add some phrases related to the limitations of the study.

Other observations:

Please improve the quality of the figures presented in the paper. Most of them are hard to read.

Please provide explanations to all the variables used in the equations. For example, the variables discussed for equation 2 cannot be found in this equation.

If possible, please try to improve the English. Thank you!

Reviewer 3 Report

Dear Editor and Authors,

This paper is a research article focused on the water dynamic within soils recovered for the reclamation of mining areas. For that, the authors ran field experiments monitoring hydrological properties of different soils. In general, the research is well conducted applying a suitable methodology and providing good results that support the discussion and conclusions. This topic has relevance at international scale so the work would be welcomed by the audience of Water. Some information is required to provide a complete description of the experiments and the presentation of the results should be improved. The authors should consider comments detailed below and revise the wording. Additional comments are annotated in the manuscript (PDF file). My recommendation is to reconsider the manuscript after major comments are addressed by the authors.

Sincerely,

Dr Daniel Ballesteros

University of Granada, Spain

Comments

The beginning of the introduction is too much focused on the study case. I suggest to start the introduction from a global point of view.

Sometimes, the authors use sometimes the term soil according to the edaphology-pedology (lines 31-32) and other times soil is employed following a geotechnical meaning (lines 38-39). Please, specify the meaning of soil.

Fig.1 would exhibit a map of China or Asia showing the study area.

Lines 114-118 present interesting data to characterise monitored soils. Please, indicate the technique/method employed to calculate the information.

Additional information about the monitored soils would be presented, as the mineralogy, the complete granulometry and full geochemical analysis.

Reviewer 4 Report

In this work, Zhang et al. proposes to investigate water movement in reconstructed soil and to evaluate the ecological or eco-toxicological effects of mining waste rock in an highly (mined) exploited arid and cold area.

 The work adds value to the current literature however minor issues require attention:

Line 92: The authors should provide GPS location coordinates

Line 97: The authors should provide source of the data stated (temp,  precipitation, evaporation)

Figure 1 (and pretty much all of them) is of extremely pour quality

Line 104: How were the field test performed? What method was used? How was the calcium oxide percentage determined, as well as SiO2, MgO and Al2O3.

Line 211: Provide a few more details regarding the neutron probe used

Line 254,261 and 323: The potential of the Fabaceae Medicago sativa was mentioned multiple times, please use citation and provide details regarding the bioremediation mechanism.

Round 2

Reviewer 1 Report

Minor comments:

  • In the Figure 4, differences in the water contents among monitoring points are hardly recognized. I recommend to separate the water content into three graphs divided by the a, b, c soil columns.

  • Related to the model calibration, clarify model calibration parameter in the manuscript. Also, provide model calibration graph to see how well your model result fitted to the observed data.

Reviewer 3 Report

The authors considered all my comments, providing persuasive answers and improving the manuscript with new data and modifications. The new version is clearly improved. My recommendation is to accept the work as the present form.

Sincerely

Dr. Daniel Ballesteros, University of Granada, Spain
